# Autonomous Flight Trajectory Control System for Drones in Smart City Traffic Management

**Dinh Dung Nguyen** **, Jozsef Rohacs * and Daniel Rohacs**

Department of Aeronautics and Naval Architecture, Budapest University of Technology and Economics,
1111 Budapest, Hungary; ddnguyen@vrht.bme.hu (D.D.N.); drohacs@vrht.bme.hu (D.R.)
* Correspondence: jrohacs@vrht.bme.hu

**Abstract:** With the exponential growth of numerous drone operations ranging from infrastructure monitoring to even package delivery services, the integration of UAS in the smart city transportation systems is an actual task that requires radically new, sustainable (safe, secure, with minimum environmental impact and life cycle cost) solutions. The primary objective of this proposed option is the definition of routes as desired and commanded trajectories and their autonomous execution. The airspace structure and fixed routes are given in the global GPS reference system with supporting GIS mapping. The concept application requires a series of further studies and solutions as drone trajectory (or corridor) following by an autonomous trajectory tracking control system, coupled with autonomous conflict detection, resolution, safe drone following, and formation flight options. The second part of the paper introduces such possible models and shows some results of their verification tests. Drones will be connected with the agency, designed trajectories to support them with factual information on trajectories and corridors. While the agency will use trajectory elements to design fixed or desired trajectories, drones may use the conventional GPS, infrared, acoustic, and visual sensors for positioning and advanced navigation. The accuracy can be improved by unique markers integrated into the infrastructure.

**Keywords:** autonomous drones; UAV; autonomous flight trajectory; inverse motion simulation; smart city integration

## 1. Introduction

Recently, science and technology are ready to develop and produce an extensive series of low-cost small remotely controlled or autonomous air vehicles as drones (generally unmanned aerial vehicles/systems—UAV, UAS, including even small pilot-less air vehicles, air taxis). The market of their civil application generated by the economy and social needs is rapidly growing. On the other hand, a severe problem blocks the rapid introduction of drones in city operations and smart city transportation. The existing air traffic management system (ATM) cannot control the predicted amount of drones operated at low altitude in the urban area between large buildings and complex environment (with, e.g., reflection), due to, e.g., (i) the limitations in the system capacity, (ii) the required workforce, (iii) the expected cost, (iv) the required duration of the system development.

To enable drones to be operated regularly as an integral part of the urban air transportation system, it is essential to develop technical solutions, formulate regulatory frameworks, and design management systems to safely conduct operations, both in the air and the ground.

The primary objective of this proposed option is the definition of routes as desired and commanded trajectories and their autonomous execution. The airspace structure and fixed routes are given in the global GPS reference system with supporting GIS mapping. The concept application requires a series of further studies and solutions as drone trajectory (or corridor) followed by an autonomous trajectory tracking control system, coupled with

autonomous conflict detection, resolution, safe drone following, and formation flight options. The second part of the paper introduces such possible models and shows some results of their verification tests.

There are three essential problems are defined that require solutions:

- definition of the flight networks, safe net of desired flight trajectories,
- developing basic traffic rules (as separation requirements) and trajectory (flight path) the following control,
- developing a series of methods, solutions for safe flight (like conflict detection and resolutions, group flights, drone following models, etc.).

The problem is more complex than in road vehicles because the drones are flying in 3D space affected by strong wind, wind flow separated from infrastructure (buildings), and air turbulence.

Generally, the real motion of drones (aircraft) is monitored as a flight path. Trajectories are predefined tubes in which the different types of drones under real disturbances may flight safely. Trajectory following control keeps the drone in trajectory tubes during its flight.

Regarding technologies and models, researchers have focused on altitude control and trajectory tracking control problems. Several scientific reports have presented the altitude control problem in the literature, such as authors in [1] used barometric to improve the altitude control performance of a quadcopter, while authors in [2] used a multi-loop PID controller, an infrared (IR) camera, and an IR beacon [3]. Besides, many studies have aimed to solve the trajectory tracking control problem by using backstepping [4], multi-loop PID controller [5], or combining a PID controller and a backstepping controller [6], sliding mode control [7], extended state observer (ESO) -based robust backstepping [8], linear quadratic regulator (LQR) [9].

Concerning the management system, given the anticipated large amounts of drones and widely varying performance characteristics, it is far beyond the capabilities of conventional Air Traffic Management (ATM) systems to deliver services for drones in a cost-effective manner. Traditional ATM framework is mainly established for human-crewed aircraft, while the absence of a pilot on-board will pose a unique set of management issues not seen in human-crewed aircraft operations, such as avoidance collision, tracking trajectories, path planning, communication, and control.

Hence, integrating drones in smart city transportation is an essential task, which requires innovative, highly automated, autonomous solutions.

Numerous universities, research institutions, high-level groups, policymakers, megaprojects deal with developing rules, methods that could support the integration of drones in the air traffic management systems [10–17].

It seems that the most promising solutions for urban air transport management must use specially structured airspaces with predefined fixed routes or fixed corridors (Figure 1) following the Singapore recommendation and [18,19] demonstrations [20]. Here corridors mean multilane "highway" channel routes.

The concept application requires a series of further studies and solutions as drone trajectory (or corridor) following by an autonomous trajectory tracking control system, coupled with autonomous conflict detection, resolution, safe drone following, and formation flight options. The second part of the paper introduces such possible models and shows some results of their verification tests. Drones will be connected with the agency, designed trajectories to support them with factual information on trajectories and corridors. While the agency will use trajectory elements to design fixed or desired trajectories, drones may use the conventional GPS, infrared, acoustic, and visual sensors for positioning and advanced navigation. The accuracy can be improved by unique markers integrated into the infrastructure.

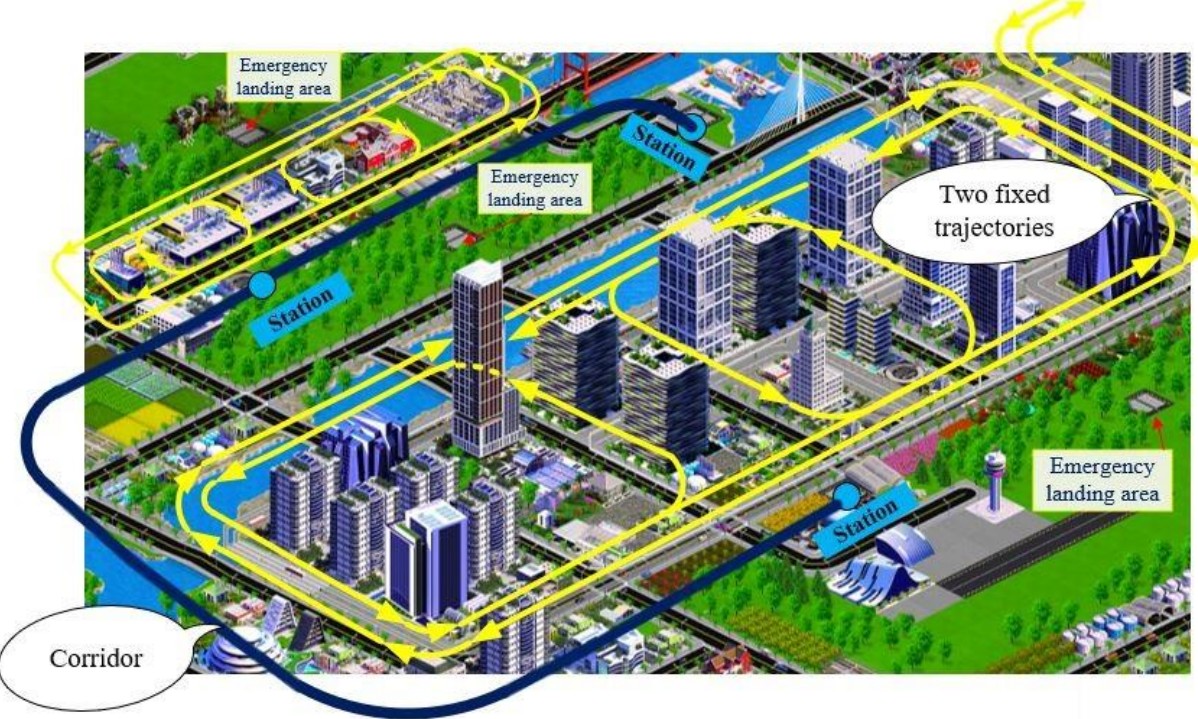

**Figure 1.** Recommended concept (following to initial idea of Kin Huat Low [19]).

The Department of Aeronautics and Naval Architecture at the Budapest University of Technology and Economics has extensive practice in developing new operational concepts as [21–24] the integration of drones in the smart city transportation systems [25–28].

Figure 2 shows the developed cockpit tool to support the EU-supported so-called Gabriel concept (magnetic levitation assisted take-off and landing concept of an undercarriage-less aircraft [29]), the primary flight data, recommended flight tunnel, and side wing profiles being displayed.

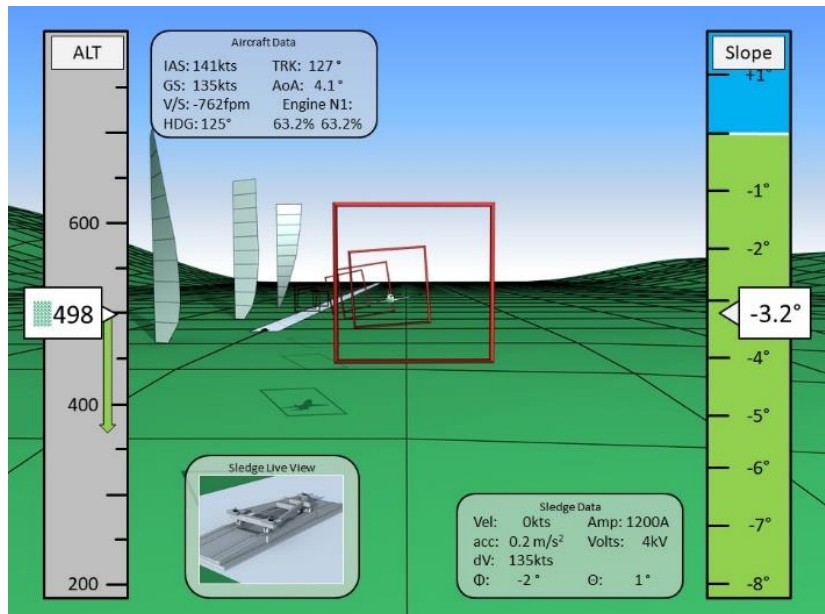

**Figure 2.** Development of cockpit tools to support precision landing.

In principle, the goal of safely managed UAV traffic is subject to a combination of essential services and control procedures. Thus, this paper's motivation is to assist the development and management of UAV operations for civil use, especially for larger drones operated at low levels.

This paper investigates the possible integration of drones in the smart city transportation system. The proposed idea is based on the "road-network" defined in 3D airspace. The drones' airway system might be constructed with typical, standard elements.

The paper has three major parts. The first part introduces the developed operational concept for drone flight operations in an urban area, including the airspace design, the recommended construction of the airways, and the essential safety requirements. It also investigates the applicable management system that must be highly automatized, with a central observation/supervision, and permit fully autonomous flights of drones using GPS and active marker systems being integrated into the infrastructure as well as defined in a GIS environment. The second part deals with the required ATM and dedicated control systems for drones or groups of drones. Drones may follow the fixed trajectories or predefined corridors. Several methods as sensor fusion, real-time GIS support, centralized dynamic sectorization, active management, fixed trajectory flowing models, predefined flight modes like coordinated turns, active conflict/obstacle detection and resolution, drone following models, formation flights should support the drone's operation in smart cities. Finally, the third part discusses the possible application, including the concept verification and validation.

The paper has two major class of novelties: (i) creating and developing the operational concept, describing the trajectory network, and their element (ii) adaptive, developing and testing some models, solutions required for supporting the application of the operational concept, including new sensor and receiver distribution including into infrastructure, developing a remarkable trajectory following model, introducing new drone following model and creating methods for improving the drone landing management.

## 2. Basic Idea—Supporting Materials

### 2.1. Operational Concept

The operational concept describes the use of the given instrument, device, machine (here drone) by targeted users [30,31]. In drone applications, the users are the operators [12,15,16,18,19,32]. In this paper, the general objective of using drones is to diversify air freight transportation, which includes the collection and final distribution of small packages directly from senders to recipients, covering thus flights in urban areas.

Drones fly by following a predefined trajectory or corridor (Figure 1). Each drone flies on its trajectory that might be part or follow the generally predefined trajectories with, for example, changing "lanes", heading, altitude, or speed. Drones can never meet other drones and drones moving in the opposite directions on their trajectory.

The fixed trajectories and corridors as airways can be classified upon the analogy of the road networks. The highways are the corridors containing several lanes in a two-way direction. The distance between the corridors and from any surrounding infrastructure/obstacle should be at least 30 m. Major or mean airways have fixed trajectories.

High-speed delivery drones will fly within a corridor connected by nodes, such as one node in the harbor area, one node in the factory area, and another in the cargo air terminal (see Figure 1). The predefined trajectory is desired for drones to avoid obstacles and each other. In addition, a safety puffer is defined, ensuring that drones cannot meet each other within any circumstance.

Before the flights, the operators must inform the drone air traffic management center of the planned flight and the expected target points. Upon other users' trajectories and surrounding data (e.g., static obstacles, minimum safe altitude), the automated center defines the trajectory for the given flight in a 3D virtual channel being optimized using the GIS map and opens a slot. Once the drone misses the open slot, the process should be reinitiated.

Today, the Global Positioning System (GPS) is the most widely used Global Navigation Satellite System (GNSS) worldwide. It provides continuous positioning and timing information globally, under any weather conditions, including constellations of satellites orbiting over the earth's surface and continuously transmitting signals that enable users to determine their position. Therefore, many scientific reports presented the developed system combined with GPS/real-time kinematic positioning (RTK) to solve the autonomous flight problem, such as [33–35]. The flight is fully autonomous, but the drone continuously estimates its position, possible conflict and adapts its motion to the real flight situation. The drones' positioning is based on GPS combined with GIS mapping [36,37], using fixed-point markers in the infrastructure and active, intelligent surveillance (Figure 3). Here GIS supports the definition of optimal trajectories based on minimizing the habitants being affected by the environmental impact or endangered by the possible emergency/accident situations.

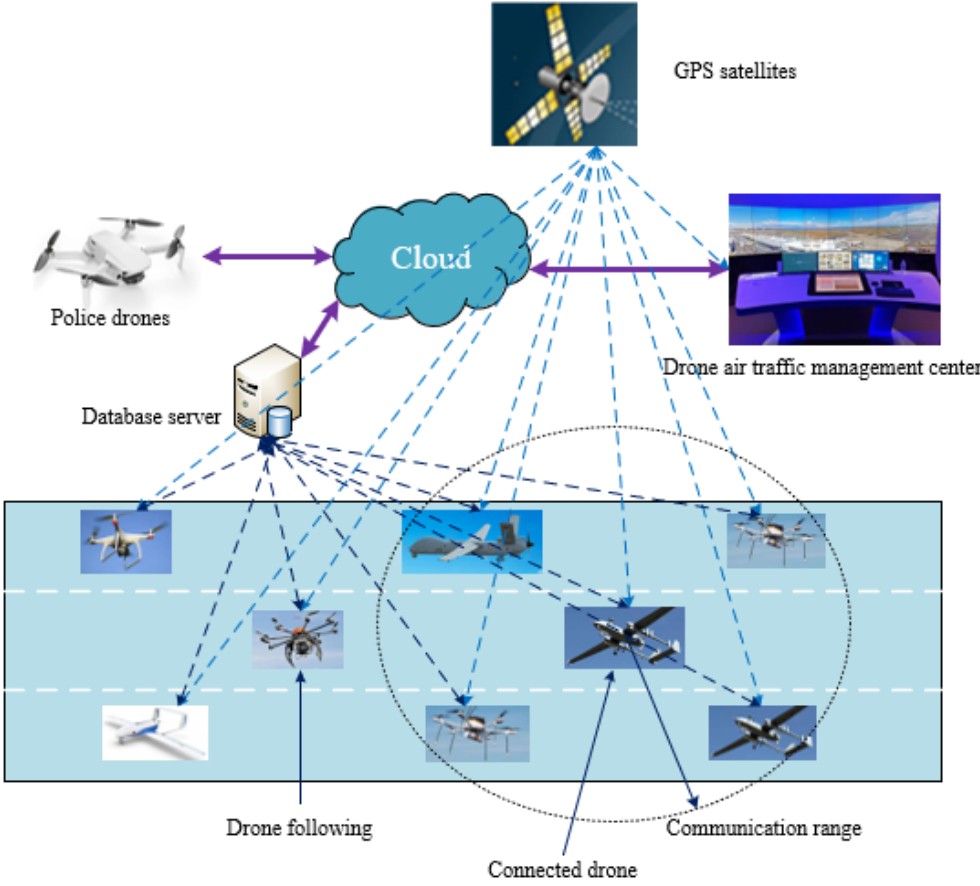

**Figure 3.** General concept and system layout of the proposed autonomous drone management system.

The general concept and system layout are shown in Figure 3.

Drones should follow the predefined trajectories/corridors. A series of safety and security solutions are used, including, e.g., safe separation, trajectory following, conflict detection, and resolution (see Sections 2.2 and 3).

The predefined trajectory might have stops, return, or round flights. After reaching the target destination or next stop, the drone might follow the next part of the predefined trajectory of "ask" a new trajectory based on the next target point. The slots for the following parts of the flight are defined and opened automatically by the operational center.

*2.2. Airway-Network*

2.2.1. Sectorization

The airway-network structure is based on an extensive study being available in the literature [18]. Airway-network is a better distribution of traffic flow that might reduce

congestion and provide more flexibility to flight schedules and routes. Therefore, it is necessary to design an airway-network to manage traffic flows better and reduce traffic congestion. Four different sectors are recommended to be used, such as geographical sector, sectors in vertical separation (between the large buildings), sectors for vertical motion (climb/descent), and sectors for restricted areas.

- Geographical sector: dedicated areas defined on the map (see Figure 4).
- Sectors in vertical separation: sectorization can be applied in the vertical direction, for example, sectors between the large buildings and above them.
- Sectors for the vertical motion: vertical moving of drones, e.g., flying up (climbing) or down (descent), can be realized as lifting of copters or flying in a spiral for fixed-wing UAVs. Therefore, a particular cylinder will define the area for changes in height (see Figure 5).
- Sectors for restricted areas can be defined for any essential reasons.

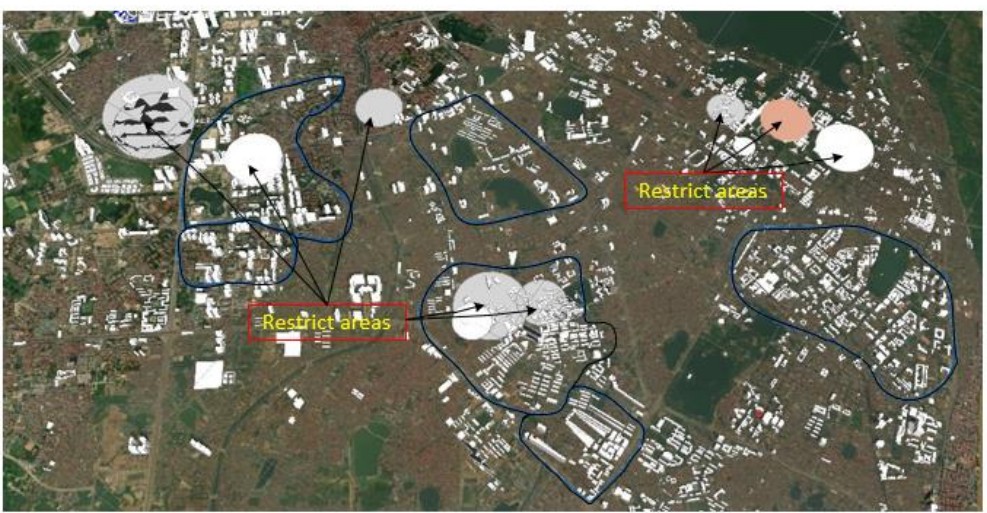

**Figure 4.** Geographical sector and restricted areas.

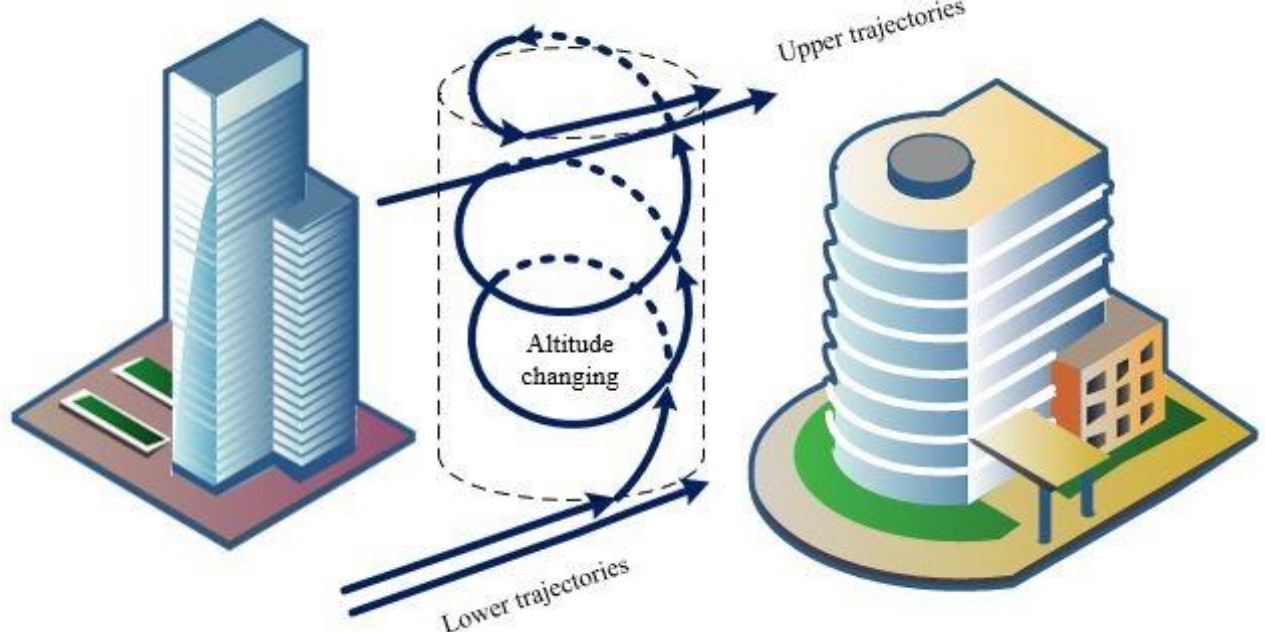

**Figure 5.** Sector for "vertical motion", changing the flight altitude.

### 2.2.2. Typical Elements

Elements of the airway-network are simple elements of trajectories, lanes in which the aircraft might fly in one stationer flight mode as a straight flight, changing the lane, descent or climb, coordinated turn (Figures 6–11).

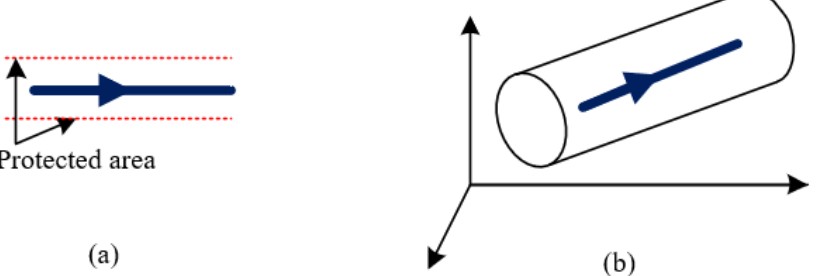

**Figure 6.** One way: (**a**) vertical view, (**b**) 3D view.

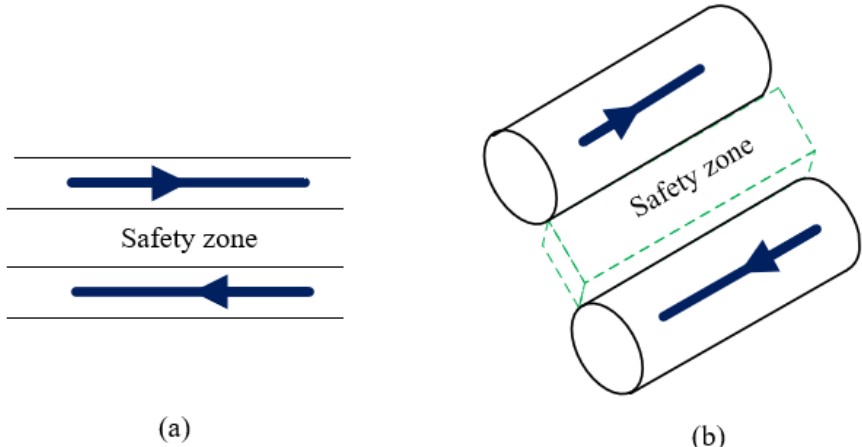

**Figure 7.** Two ways: (**a**) vertical view, (**b**) 3D view.

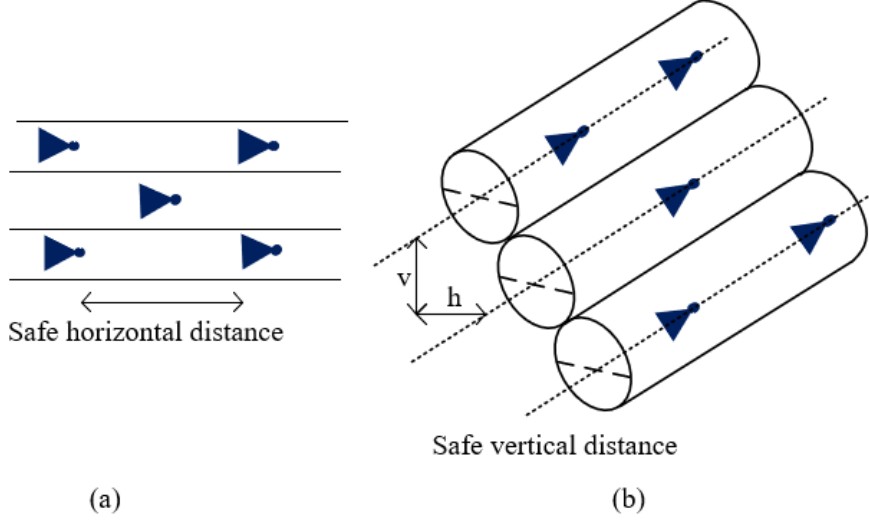

**Figure 8.** Multi-lanes in one direction: (**a**) vertical view, (**b**) 3D view: v—vertical safe distance, h—horizontal safe distance

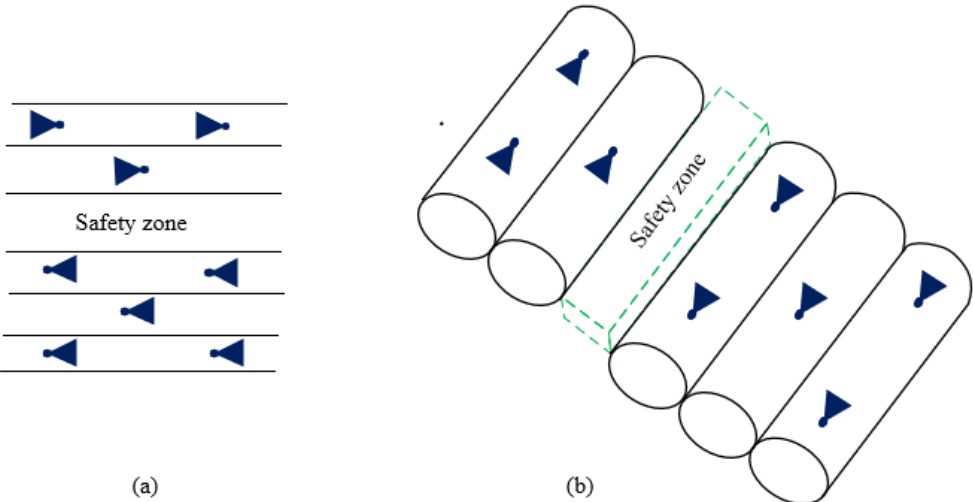

**Figure 9.** Multi-lanes in two ways: (**a**) vertical view, (**b**) 3D view.

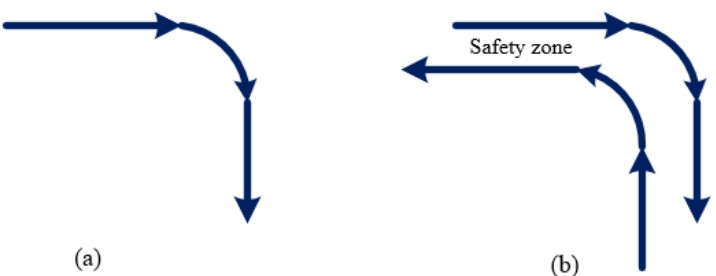

**Figure 10.** Turning: (**a**) in one way at the same altitude, (**b**) in two ways at the same height.

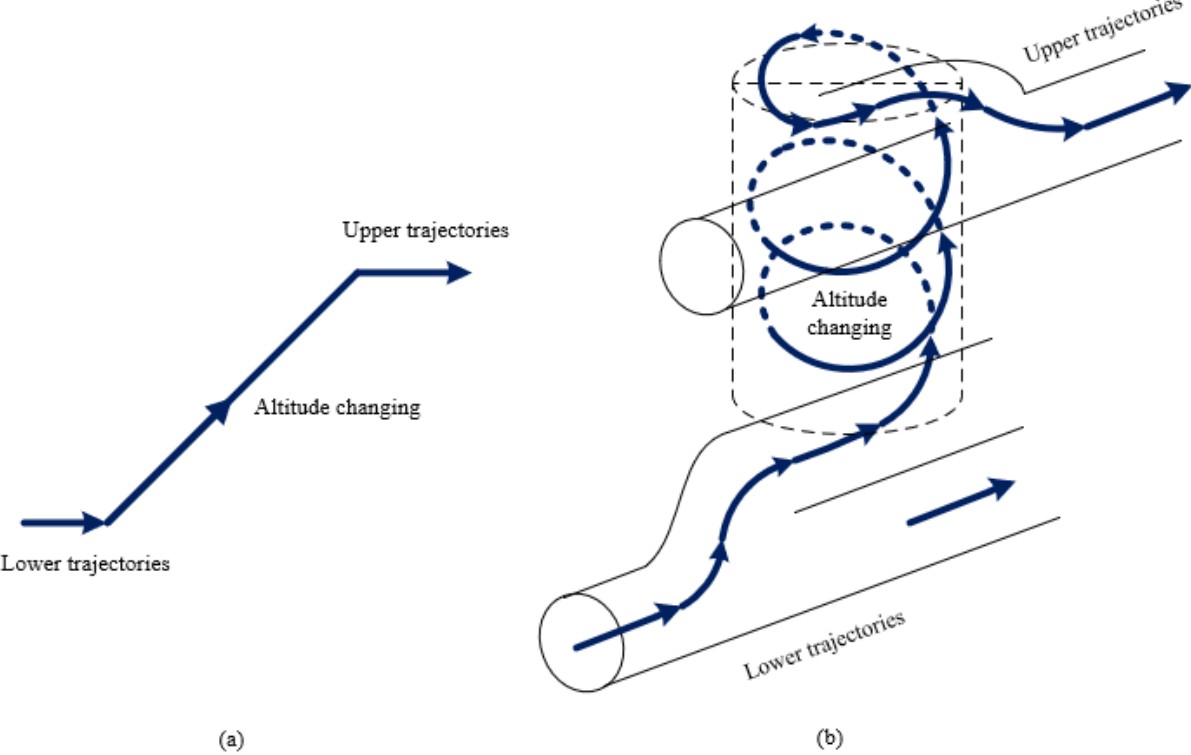

**Figure 11.** Changing altitude in the same direction: (**a**) with a straight flight, (**b**) with coordinated turn.

There are two different crossing options: (i) the straight-line crossing with no heading modification after the crossing (changing lane) (Figure 12a) and (ii) the crossing with heading modification, including possibly a vertical motion due to the modified heading (Figure 12b).

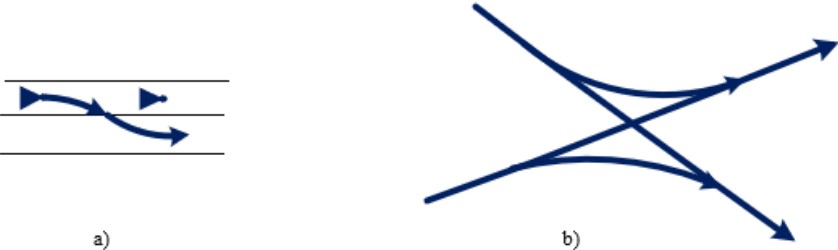

**Figure 12.** Crossing: (**a**) changing lane, (**b**) changing heading (in top view).

However, changing the direction (or heading) is not as straightforward. To minimize the number of potential conflicts, lanes of different headings are at different altitudes. Thus, heading modifications lead to the following six simple maneuvers, as shown in Figure 13.

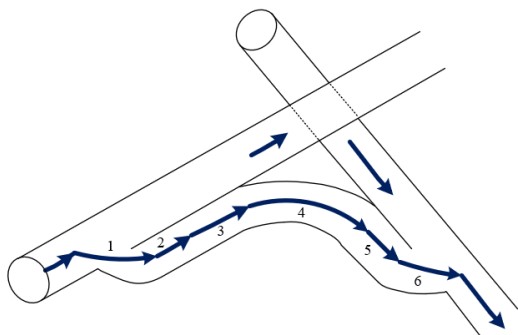

**Figure 13.** Changing heading at different altitude: 1—changing to a new lane, 2—flying in the new lane, 3—increasing/decreasing the altitude, 4—turning on the same altitude, 5—flying at the new lane in the desired heading, 6—merging in the lane at the same altitude and at the desired new heading.

### 2.3. Safety and Security Aspects

There are numerous regulations [12,14,38–40] and related works [41–48] focusing on drones' safety and security aspects. In this paper, some recommendations are proposed to facilitate the integration of drones in the urban air traffic system. The general idea applied to developing the recommendations is to have at least one second for reaction in the worst case.

### 2.3.1. Safety Rules Applied to the Definition Airways

To define the airways, the authors investigated and evaluated several recent regulations and related works focusing on drones' safety and security aspects. The following assumptions were made as a means to define the airway network under study and the research scope:

- Defining speed limits to 30 m/s for the corridors, 20 m/s for drones flying in fixed trajectories at a minimum of 20 m from any infrastructures (buildings), and 10 m/s for drones moving 20 m closer (but 5 m away) from the infrastructure.
- The drone's recommended longitudinal separation in a fixed trajectory depends on the speed, difference in speeds, and the level of cooperation between the given drones. Preliminary longitudinal separation "time" should be a minimum of one second plus an additional second by each 10 m/s of flight speed sec, for the non-cooperative

vehicles that should be increased when the follower (*i* + 1) drone has a greater speed, $\Delta v$ (m/s) being compared to the leading *(i*-th) drone for $\Delta v/3$ in sec. In the case of cooperative drones, the longitudinal separation time can be decreased by 30–40% (depending on the actual intensity of air turbulence), and for the case of formation flight, another 30%.

- The lateral separation (horizontal and vertical direction) is defined by Figures representing the recommended typical elements of airways. As a general rule, the horizontal and vertical distance between the drones' center of gravity heading in the same direction should be equal to 5–8 times their maximum dimensions. If drones fly in the opposite direction, a particular safe distance equal to an empty lane should be applied.

- The airways and the total network should be composed of the elements described above (Section 2.2), and the drones might change lanes in the horizontal or vertical direction only.

- The defined trajectory as a channel for the given drone is fixed and cannot cross any other trajectory.

These rules were suggested as a means to eliminate or mitigate the causal factors or unsafe situations. Although this study proposes the safety requirements related to the drones, we derived requirements assigned to the aviation authority and the manufacturer. However, a complete analysis of the authority and manufacturer levels was beyond the scope of this study. Thus, an analysis of the control actions, their unsafe states, and the corresponding causal factors and scenarios must be studied and developed to improve the effectiveness of these requirements.

### 2.3.2. Safe Airspace, Airway-Network Design

Sectorization should be applied as a primary approach being developed and implemented by the drone air traffic management center.

Different 3D sectors (in size) are used depending on the geographical aspects, ground obstacles (provided by GIS), predicted market needs, demand for drone services, and expected traffic intensity. Sectors can also be different in the vertical dimension: lower levels in the urban areas might contain more sectors (to facilitate the various operations) than upper levels above the city.

The sectorization is dynamic and active, which means that the sectorization might be changed dynamically depending on the actual historical/predicted data and act upon the real measured actual situations.

Trajectories and, generally, airway-networks are developed and designed using multi-disciplinary and multi-objective optimization to minimize the total impact of drones with minimum total cost. Here total means the sum of all the effects or costs of all drones. Impact includes all the immediate, short, and long term effects and externalities caused by drone operations, such as the impact on nature, built and living environment, health problems initiated by the emissions and accidents, effects on the economy. Total cost is determined by taking into account all the costs, including, for example, the operation, production, the development and operation of all the required infrastructures, or the external cost affected by accidents.

The airway-network is operated in urban areas, where accurate positioning and traffic management require special supporting rules and a built environment. Rules might be developed by the partial implementation of the road traffic rules (including even the road and traffic signs) and unique markers being integrated into the city infrastructure (see Figure 3).

The airway-network (as the sectorization) is operated by passive, dynamic, and active methods (see Section 2.4).

Airway-network should have special zones for an emergency landing if safety and security problems are detected (see Figure 1).

### 2.3.3. System Operation

The integration of drone flights in the air traffic management system is a real challenge [49]. The largest and most important part of the safety aspects related to the drone is the drone safety system composed of a hierarchical system, air traffic management (see Section 2.4), and methods supporting safe drone flights in the airway-network (see Section 3. Methods).

### 2.3.4. Security Aspects

This paper deals with the civil and commercial application of drones in urban areas—smart cities. The recommended urban drone traffic system and management might operate relatively large (up to 1600 kg take-off mass as mid-size cars) autonomous vehicles in corridors and relatively small air vehicles (mostly with less than 60 kg take-off mass) following trajectories/channels. The corridors are far enough from the built environment to be able to react even when drones are unintentionally leaving the fixed corridors (e.g., due to malfunction). The smaller drones following the fixed trajectory may cause fewer damages and problems, limiting the possible unlawful actions.

In such an environment, four major security problems should be solved:

- Cybersecurity: as a general problem of highly automated and autonomous vehicles; objects having large and centralized info-communication and management systems;
- Using drones as weapons for unlawful actions;
- Flight into restricted areas;
- Attack on drones using arms, guns, weapons.

These problems might be solved by implementing the available and emerging security methods and developing a closed system for the drones' traffic management. The latter means that all operators, service providers, and drones should be integrated into one system. The major methods and solutions being recommended are the following:

- primary (passive) surveillance: using fixed optical and microwave systems, sensors, receivers of which are integrated into the urban environment along the fixed trajectories, channels, corridors and using larger fixed surveillance radars and mobile drones for further detections (of drone flights);
- secondary (active) surveillance: developing and implementing mini transponders that might cooperate with the surveillance system within low distance, up to 600 m. The system elements should be integrated in the urban area along the fixed trajectories, channels, corridors;
- secure communication system: as it introduced, internet/cloud-based with particular security protocol using continuously changing coding system and the drone's security identification being able to detect possible anomalies in the communication or potential cyber-attacks;
- onboard security controller—first level: a unique device that avoids to enter in restricted areas;
- onboard security controller—second level: devices that detect any security problems, attacks, and initiates the forced landing of the drone on the nearest emergency landing area;
- defense and protection system: that, as part of the total drone traffic management system, automatically detects the possible violation of the defense zone to attack/intercept/destroy the detected failed or unlawfully flying drones.

### 2.4. System Definition

A drone management system is a sub-system of the overall unmanned traffic management system (UTM) that guarantees safe, environmentally friendly, effective, and sustainable mobility in urban/city areas. It is a sub-system only, but it has fast interconnections with air traffic management systems, including, e.g., sharing airspace, conflict/obstacle detection and resolution, and optimal trajectory control. Nowadays, new forms of transportation, such as urban air transportation (drones, air taxis), must be integrated in the total

transportation management (see Figure 14 [25]) and harmonized with other autonomous vehicles [50].

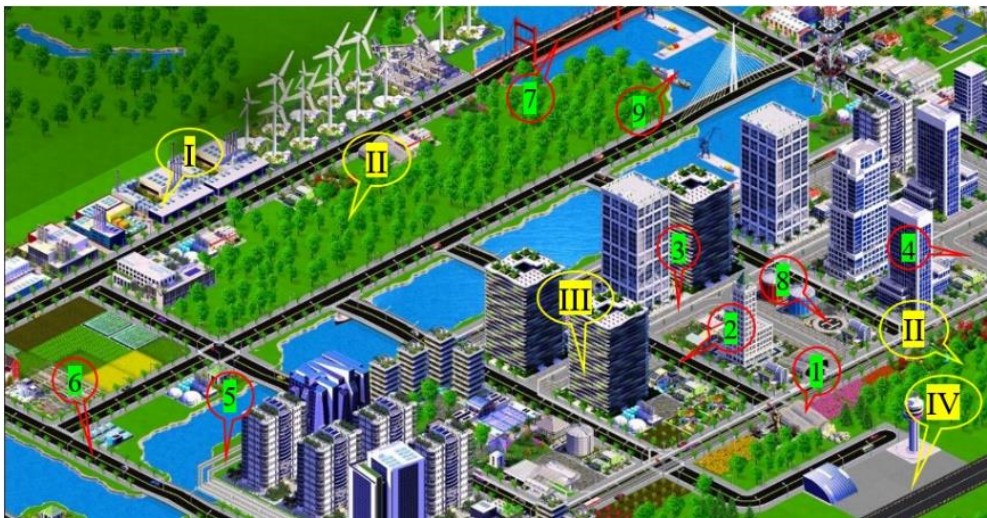

**Figure 14.** Urban total transportation system: I—industrial area (factories), II—Forest area, III—urban area, IV—airport area, 1—underground, 2—road, 3—upper ground, 4—path, 5—railway, 6—highway, 7—freight transport, 8—urban air transport, 9—water transport.

In the future, with the introduction of autonomous/connected drones and highly automated traffic management systems, the operator roles will be shifted from active control to passive observation and only expected to take active control in case of exceptional or emergency cases. Such a control environment should be supported with adequate data to permit the UTM to support the smart city vision.

The drone management is based on a hierarchical concept distinguish following six categories:

- Non detected objects: that does not appear on the surveillance screen;
- Detected objects: that appear on the surveillance screen, but it is unknown whether it is passive, non-cooperating, or shows non-relevant target, such as birds;
- Semi-active or simple cooperating objects: that provide at least some information to the operation center;
- Active or cooperating objects, or service providers: that report information on the objects operating in the city, the available information should contain data on the type of the vehicle, its identification number, load the instantaneous position, purpose, and final destination;
- Connecting vehicles that cooperate together and harmonize their movements passively or actively, e.g., moving in formation, or using conflict detection and resolution based on the exchanged information;
- Contract-based drones that have some preferences, which is contract-based and must pay for the service provided.

The total drone transport management system can be made as shown in the Figure 3. There are unique markers that can be integrated in the urban built areas, as the walls of the buildings, roads (especially in road crossing areas), supporting further and more accurate positioning. Simultaneously, surveillance and security detection sensors can also be integrated in the urban environment.

## 3. Methods

The second part of this paper deals with the required air traffic management and flight control for drones or a group of drones. The drones might follow fixed trajectories or predefined corridors. Several methods as sensor fusion, real-time GIS support, centralized

dynamic sectorization, active management, fixed trajectory flowing models, predefined flight modes like coordinated turns, active conflict/obstacle detection and resolution, flight drone following models, formation flights should support the drone's operation in smart cities.

As it can be understood from the previously described basic idea, the proposed system synthesized the air traffic, air traffic management practice with the urban road and interurban highway transports' solutions, including the traffic rules, traffic organization, and control. Drones moving in urban areas might often fly at a relatively low altitude and between buildings in the air with intensive air turbulence partly caused by wind (flow) separation from infrastructures (buildings). Therefore, some applicable automation methods like vehicle following models might be significantly varied and different in road and drone traffic. Therefore, one of the most challenging autonomous drone control problems is a flight through a narrow gap. It requires the drone to pass through the center with its attitude aligned with the gap's orientation, minimizing collision risk. One solution for this problem is the one-by-one following drone process in "tight tunnels" between the houses without a traditional planning and control pipeline. Hereafter, some information about the applicable supporting methods tested by authors in simulations and partly in an experimental physical environment are presented.

### 3.1. Sensor Fusion Tools in Support of Autonomous System

Autonomous systems rely on a sensor suite that provides data about themselves and sensors that sense the environment. In many cases, the autonomous system uses commercially available sensors. For pose estimation, sensors commonly used are accelerometers, gyroscopes, magnetometers, altimeters, or GPS. Sensors that sense the environment include radars and vision cameras, which provide detections of objects in their field of view. Lidars provide point clouds of returns from obstacles in the environment, and in some cases, ultrasound and sonar sensors. Some signal processing occurs in the sensor system, often including detection, segmentation, labeling, classification, and in some cases, basic tracking to reduce false alarms.

Authors in [51] introduced the requirements and a possible solution for a set of tools in the perception stage in which inputs from various sensors are fused to provide a single estimation of the environment around the autonomous system. This toolset integrates sensor fusion and tracking algorithms into complete tracks, simulating sensor data, swapping trackers, testing various fusion architectures, or evaluating the overall tracking results.

Authors in [52] presented a work that focused on feasibility testing of ground Electro-Optical/Infrared (EO/IR) and characterization of acoustic viability with the intent to expand nodes in a wireless communication network of nodes in a Drone Net Sensor Network in an air-column of 1 km diameter. In this study, the active sensors, RADAR and LIDAR, were used on the ground as a secondary validation. In contrast, passive sensors competed with RADAR in terms of classification and identification performance at a lower cost.

### 3.2. Desired Trajectory Following Management

Several investigations deal with flight planning, including the uncertainties in the environment, precise aggressive maneuvers, or low altitude flights [53–55]. The trajectory tracking or path following, especially in quadrotors, could be supported with backstepping trajectory control [8], barometric altitude measurement fault diagnosis method using feedforward neural networks [1], multi-loop PID controller [2], infrared (IR) camera and IR beacon [3], feedback linearization control-oriented algorithms, non-linear guidance law, or carrot-chasing geometric algorithms [56].

In the proposed airway-network system, drones should follow the fixed trajectories, channels, or corridors. In the environment, the drones' actual position and motion can be measured with sensors being integrated in the urban areas/infrastructure.

In a simplified case, the trajectory following methods might be based on a discrete time-variant state-space representation of the drone motion:

$$\mathbf{x}[k+1] = \mathbf{A}(\mathbf{x}[k], \mathbf{z}[k], T)\mathbf{x}[k] + \mathbf{B}(\mathbf{x}[k], \mathbf{z}[k], T)\mathbf{u}[k], \tag{1}$$

$$\mathbf{y}[k] = \mathbf{C}(\mathbf{x}[k], \mathbf{z}[k], T)\mathbf{x}[k] + \mathbf{D}(\mathbf{x}[k], \mathbf{z}[k], T)\mathbf{u}[k], \tag{2}$$

where $\mathbf{x}$, $\mathbf{y}$, $\mathbf{u}$, $\mathbf{z}$ are the state, output, input (control), and environmental vectors, $\mathbf{A}$, $\mathbf{B}$, $\mathbf{C}$, $\mathbf{D}$ are the state (or system), control, output and feedthrough (or feedforward) matrices, and k is the time variable.

The drone's trajectory, position in the 3D space—in the coordinate system being connected to the drone management center—could be determined from the state characteristics, and more precisely from the velocity components. Similarly, the measured actual trajectory characteristics can be transferred into the changes in state vector elements. Therefore, in the following, the Equation (1) will be used only.

The trajectory following model is based on a unique inversion-model-based control. It is not a simple feedforward control, as the predicted trajectory is based on approximation and interpolation of the controlled drone.

Let suppose the $\mathbf{u}_d$ as desired control keeps the drone on the trajectory, and the series of previously measured $\mathbf{x}[k]$, $\mathbf{z}[k]$, $\mathbf{y}[k]$ are available. The actual matrices $\mathbf{A}$, $\mathbf{B}$, $\mathbf{C}$, $\mathbf{D}$ can be defined by the state and environmental vectors, $\mathbf{x}[k]$, $\mathbf{z}[k]$, while the state and output vectors, $\mathbf{x}[k]$, $\mathbf{y}[k]$ can be used to determine the prediction of the future state vector, $\mathbf{x}_p[k+1]$. The difference in the predicted and the desired trajectory characteristics can be used to determine the required changes in the desired input to return the drone to the predefined fixed trajectory. Applying this approach to the Equation (1), leads to the followings:

$$\Delta\mathbf{x}[k+1] = \mathbf{x}_p[k+1] - \mathbf{x}_d[k+1] \tag{3}$$

$$\Delta\mathbf{u}[k] = (\Delta\mathbf{x}[k+1] - \mathbf{A}(\mathbf{x}[k], \mathbf{z}[k], T)\mathbf{x}[k])(\mathbf{B}(\mathbf{x}[k], \mathbf{z}[k], T)\mathbf{u}[k])^{-1} \tag{4}$$

and

The application of the method is demonstrated in the Figure 15.

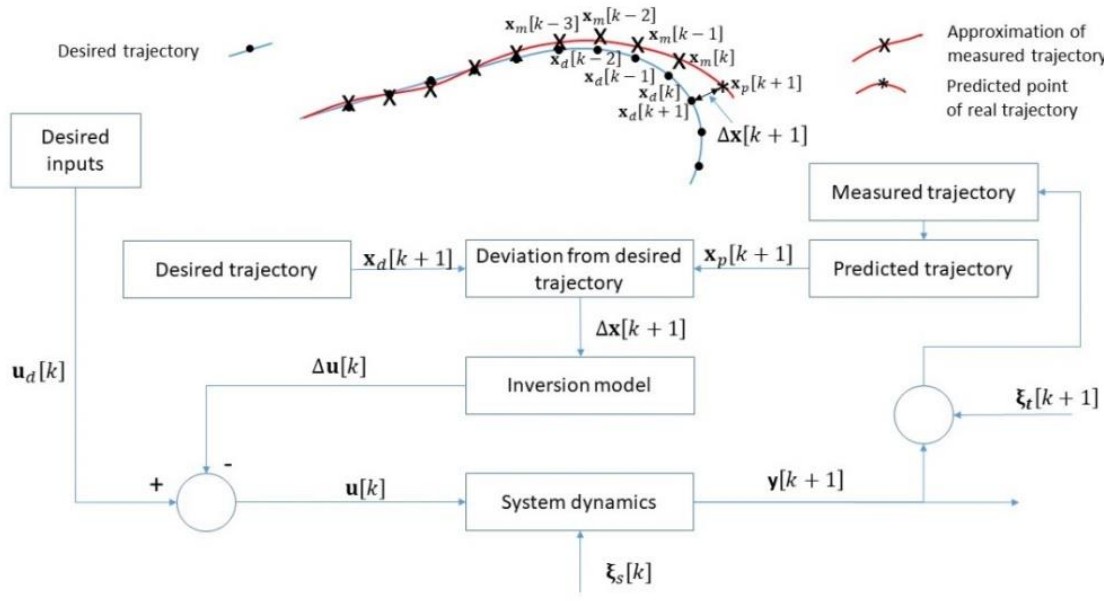

**Figure 15.** The proposed trajectory following model.

The proposed method tested in simulations demonstrated its applicability even in the 5th level of wind (8.5–10.5 m/s on Beaufort scale), inducing strongly separated vortices and

cases when the GNSS essential positioning may have difficulties (in urban areas, between the large houses).

### 3.3. Following Process

Generally, UAVs follow a reference trajectory to do their task. The reference trajectories are usually planned as straight lines, curves, or a combination of both. To achieve an excellent autonomous flight, a precise, robust, and effective trajectory-following guidance law is required. Various guidance laws were developed for trajectory following, including waypoint following [57], vector-field-based trajectory-following guidance laws [58], adaptive optimal trajectory-following guidance law using the linear quadratic regulator technique [59], nonlinear guidance law [60] virtual-target-based approaches [61], or PID controller techniques [62]. The authors in the study [63] presented new trajectory-following guidance for a UAV to track a virtual target moving along the reference trajectory based on a line-of-sight (LOS) angle constraint. This approach addressed large initial heading angle errors with satisfactory performance. A problem of coordinated path following for fixed-wing UAVs with speed constraints in the 2D plane was investigated [64]. In this study, a hybrid control law based on an invariant set to solve the coordinated path following problem of a group of fixed-wing UAVs. However, when the UAVs are outside the coordination set, collision avoidance is not guaranteed because they execute the single-agent level control law.

One of the areas that drones have a high potential in contributing to their involvement in the concept of smart cities. With the advances in big data and the Internet of Things (IoT), researchers have also referred to the previously mentioned concept of a swarm of UAVs to use for urban sensing around the city. However, it should be noted that except for the technical issues that this approach may raise, e.g., collision avoidance systems, navigation, the way this system of drones collaborates and becomes an effective means of collecting accurate and massive information is a complex optimization problem.

When the number of drones increases, severe accidents can appear in the sky, even in simple situations. The investigation of drone traffic safety and the intelligent transportation system's development requires drone-following models describing one-by-one following process in the traffic flows. The drone-following models are based on the idea that each drone can be flown under its leader, expressed by the function of safety distance or relative velocity of two drones. For example, if three drones are flying on the same route simultaneously, two of them can fly following the leader.

The basic and probably most used car-following model was developed by Gazis et al. [36] based on keeping the safe distance according to the relative distance. This model is often called as SD (safe distance or system dynamic) model. In the drone following process, the drone's velocity depends on the traffic situation, namely on the distance to the drone ahead and its velocity. This approach led to the linear models assuming that its controller controls the drone's acceleration to keep zero relative velocity to the drone ahead.

The SD model is given as follows:

$$\ddot{X}_n(t+T) = \lambda \frac{\left[\dot{X}_n(t)\right]^p}{\left[X_{n-1}(t) - X_n(t)\right]^q} \left[\dot{X}_{n-1}(t) - \dot{X}_n(t)\right] \tag{5}$$

where, $X_n(t+T)$—the acceleration of $n$-th drone after a reaction;

$X_{n-1}(t) - X_n(t)$—relative distance between the $(n-1)$-th drone and the $n$-th drone;

$\dot{X}_{n-1}(t) - \dot{X}_n(t)$—relative velocity of $(n-1)$-th to the $n$-th drones in time $t$;

$T$—delay time of a controller;

$\lambda$—a weight coefficient related to the controllers;

$p, q$—parameters related to velocity and distance of the drone ahead.

It seems this model is well applicable to the drones flying in the desired flight path. However, the air turbulences and wind flow separated from infrastructure cause rather stochastically disturbed motion of drones. With the characteristics of advanced controllers,

the controller's relative distance and actual reaction time are added to the control close-loop. This approach leads to an improved model, called the Markov model.

The Markov model is based on the approximation of the stochastic process of velocity decision. One advantage over the SD model is that the inputs of the controller are different velocities and deviations in relative distance between the drones, which can be described such as follows:

$$\ddot{X}_n[\text{k}+1] = c_v\left(\dot{X}_{n-1}[k] - \dot{X}_n[k]\right) + c_x\left[(X_{n-1}[k] - X_n[k]) - \Delta X_{pdn}\right] + \varepsilon[k] \qquad (6)$$

where, $c_v$ and $c_x$—coefficients depending on the time, given drone and controllers;

$\Delta X_{pdn} = \dot{X}(t)$—the predefined safety distance between the drones;
$k$—the number of steps in a chain ( $t = k.\Delta t$);
$\varepsilon[k]$—the random value disturbing the process.

### 3.4. Obstacle Avoidance Method

Along with drone applications' spreading trend, drone collision's flight safety with buildings, helicopters, and the landscape becomes an urgent issue for civil and defense agencies. A collision avoidance system is necessary for drone flights, especially for autonomous drones in dense airspace shared with other aircraft to guarantee airspace security. Conflict detection and collision avoidance is also a valuable tool for highly automated and autonomous vehicles. There are several simulation systems for algorithms tested and designed in laboratories. The obstacle model is one of the critical parts of these systems, described as the following. Assume that each obstacle is prescribed in a cylinder with the center $C_{Bl}$ and radius $r_{Bl}$, as shown in the following Figure 16. The surfaces of cylinders can then be used to form constraints for obstacle avoidance. Accurately, the safe distance $d_{s,l}$ from the obstacle $l$ is calculated from the cylinder center to its surface at the flying height.

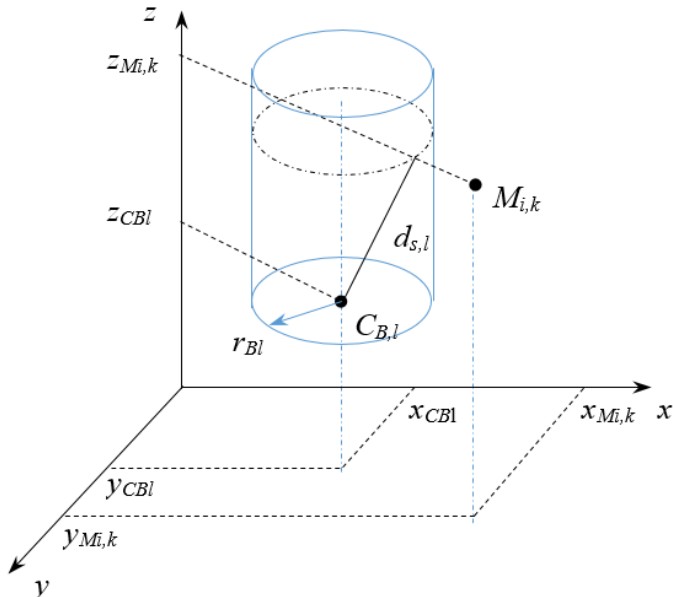

**Figure 16.** Obstacle representation and safe distance calculation.

### 3.5. Desired Landing Orbit for UAVs

The landing approach is one of the critical stages of the entire flight to bring the UAV to land safely at the desired location. Common landing approaches consist of the following stages: (i) heading against the direction of the wind, (ii) descending, (iii) slowing down. However, this process will be influenced by several factors such as wind disturbance, general aerodynamic force, the traction force of an engine, and the propeller's reaction moment.

Methodologies used to determine and calculate the landing areas are based on solving the aircraft's motion equations and analytical methods. Based on the landing areas, the desired landing orbit is estimated, within the UAV can land accurately at the desired position.

The UAV landing areas include the following three zones (Figure 17):

- Deceleration zone: this is the smallest circle on the horizontal plane containing the projection of the UAV's orbit, which flies straight with the decreasing speed during the landing approach. Then, the deceleration zone's shape is a circle with a center 0 and radius $R_1$;

- Descending zone: this is the smallest circle on the horizontal plane containing the projection of the UAV's orbit, which flies in the process of altitude reduction. This area is a circle with a center 0 and radius $R_2$;

- Directive zone: this is the smallest circle in the horizontal plane containing projections of two circles with radius $R_{min}$. Two circles tangent to each other at the opposite of the wind direction.

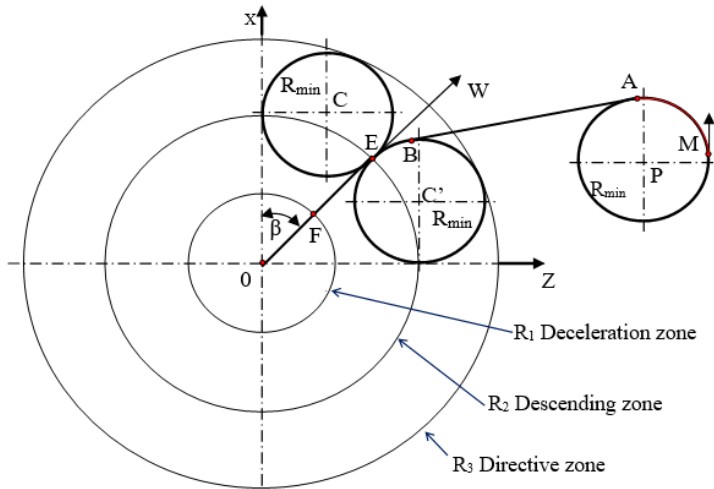

**Figure 17.** The proposed UAV landing zones.

The $R_{min}$ is the smallest rounding radius of the UAV. Thus, the directive zone is the circle with center 0 and the radius $R_3$.

UAVs' landing trajectories generally consist of approach, glideslope, and flare. A successful landing would depend on the selection of the landing trajectory and its implementation.

UAVs' landing processes consist of three stages: the directive stage, the descending stage, and the deceleration stage. These stages are determined when the UAV is into each landing zones. Landing zones will be determined by knowing the radius of each region. The most common method is to investigate UAVs' kinetic dynamics by solving the differential motion system. Therefore, UAV dynamics will be used to calculate the deceleration zone, and then the remaining landing areas will be identified by analytical methods.

## 4. Results

This part discusses the possible application, including the concept verification and validation.

### 4.1. Drone-Following Process in the Traffic Flow

Drone-following models for managing drones in smart cities' transportation management systems were introduced in previous studies [65,66]. Such models were based on the initial idea that drones fly towards a leading drone in the traffic flow. This approach has been a novel method for managing a group of drones in smart cities.

This sub-section provides the main results obtained in the simulation experiments on the SD and Markov models (see Figures 18 and 19).

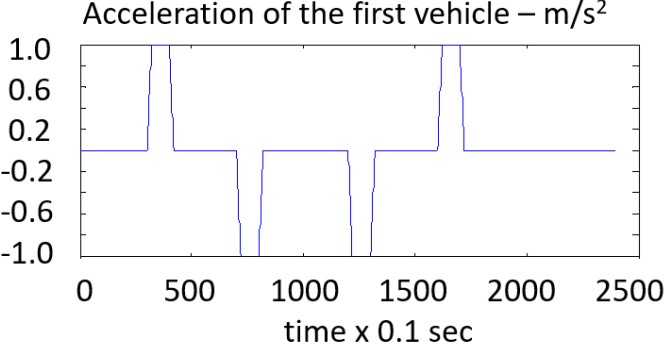

**Figure 18.** Acceleration, deceleration of the first drone applied in verification tests.

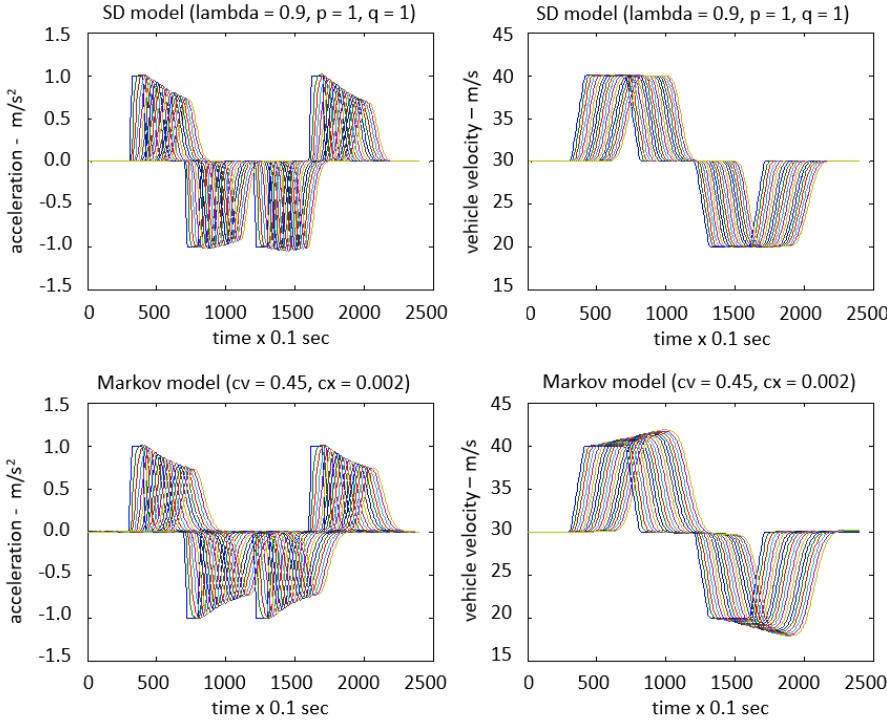

**Figure 19.** Verification results for comparison of the SD and Markov drone following models.

It can be noted that there is no accident and no unrealistic deceleration. The velocity of the followed drone is changed according to the speed of the drone ahead. However, the followed drone can react quickly compared to the leading drone's reaction because of the difference in its acceleration.

Even though the SD and the Markov models are quite similar, the followed drone's reaction in the SD model is earlier than that in the Markov model. Besides, the motion of the followed drone indicates that the stable state is slower in both models. In comparison with the SD model, the Markov model considers the changes in relative distance between drones. Moreover, the more significant the variation of the followed drone's velocity in the Markov model, the much smaller the relative distance is between the drones.

These results verified that the developed Markov model might perform the longitudinal safety separation of drones as the SD model. In general form, the results partly validated the proposed method that is part of research aiming to integrate drones with the urban intelligent transport system. The drone-following method needs to verify by several immediate next steps, such as the safe distance being measured in the drone directly in front and two drones beside, designing and conducting an experimental study to collect quantitative information regarding drone performance in space.

As it seems, the developing Markov model might be more accurate in case of motion of drones in significant air turbulence and separated wind flow from the infrastructure, and it can be used in areas where problems with GPS positioning might have appeared, especially comparing and working together with the GPS techniques or acoustic sensors, etc.

### 4.2. Experiment Results of Drone Management System

In experimental studies, a cloud-based managing method had been applied to drones flying in a smart city environment [27], including the physical, cloud, and control layers.

The experimental result is demonstrated in the Figure 20. Initially, the drone was placed at a home position. When a drone received the GCS command, it took off and did a mission, visited the created waypoints. The results show that the desired trajectory and actual trajectory are correlated. The gap between the two trajectories represents GPS location because the drone receives the GPS location.

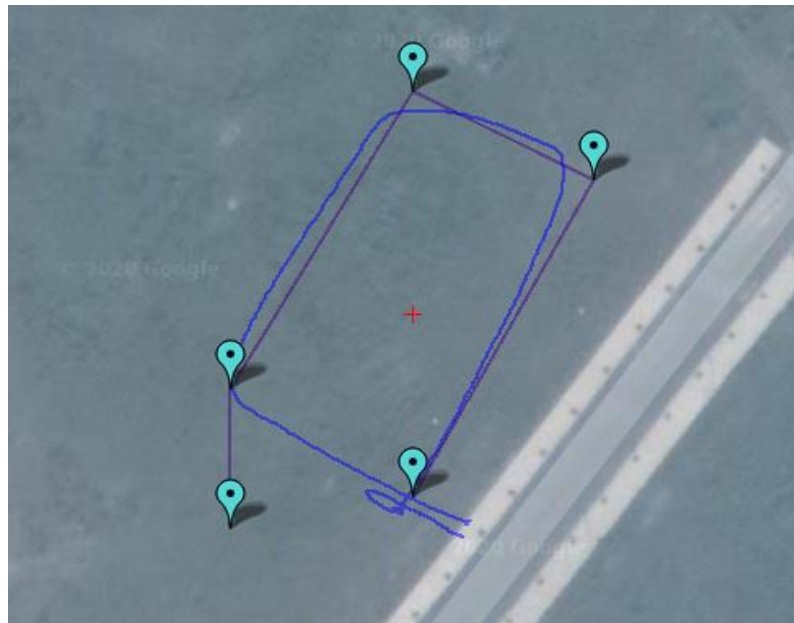

**Figure 20.** The difference between desired and real trajectories (pink line—desired trajectory, blue line—real trajectory).

It can be observed that proportional-integral-derivative (PID) control is entirely sufficient to follow set positions at low speeds. (When the drone aerodynamics changes slightly and s no wind disturbance). For initial test flights, proper tracking was achieved, even in case of moderate errors.

It is illustrated in the Figure 21 that a linear controller achieved the altitude flight control with the increase in the vertical speedup. The Flight Control Unit (FCU) has performed extremely useful in height control despite a minor gap between the desired and actual altitudes.

During the experiments, the drone's video streamed downwards, facing the camera, allowing us to observe and control the drone in a real-time environment.

These experimental results demonstrated that the proposed and applied CbDMS (cloud-based drone managing system) is a cloud solution that enables drones' management and control in a real-time environment. The monitoring efficiency can be increased by raising the regularity of refreshing GPS coordinates or adding filtering techniques (Kalman filters).

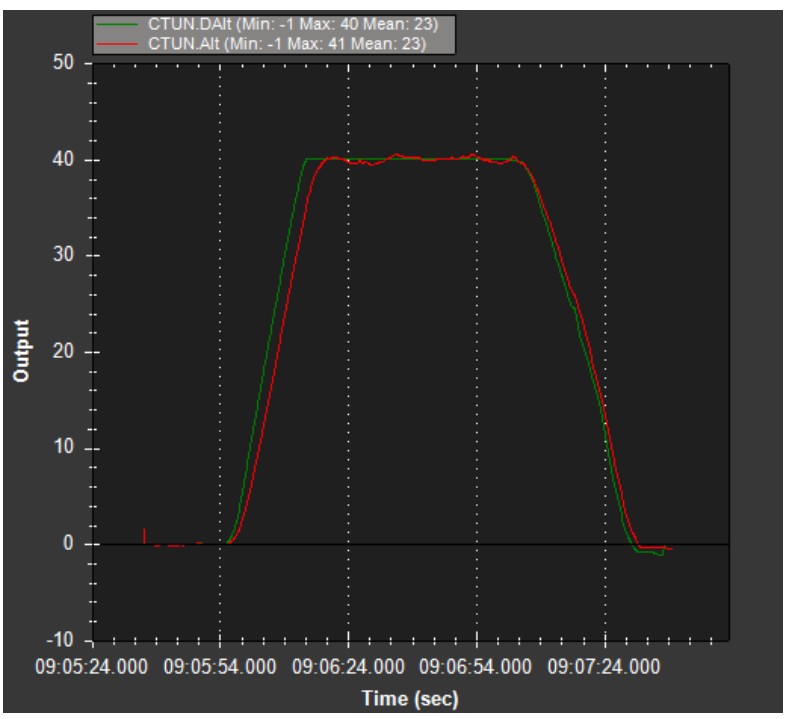

**Figure 21.** The difference between desired and actual altitude of drone, green line—desired altitude, red line—actual altitude.

### 4.3. Calculating the Desired Landing Orbits for UAVs

The previous research presented the methodologies using to determine and calculate the landing stages [67]. Based on the landing areas, the desired landing orbit is estimated, within the UAV can land accurately at the desired position. The simulation results for UAV landing in the given direction are shown in the Figures 22 and 23.

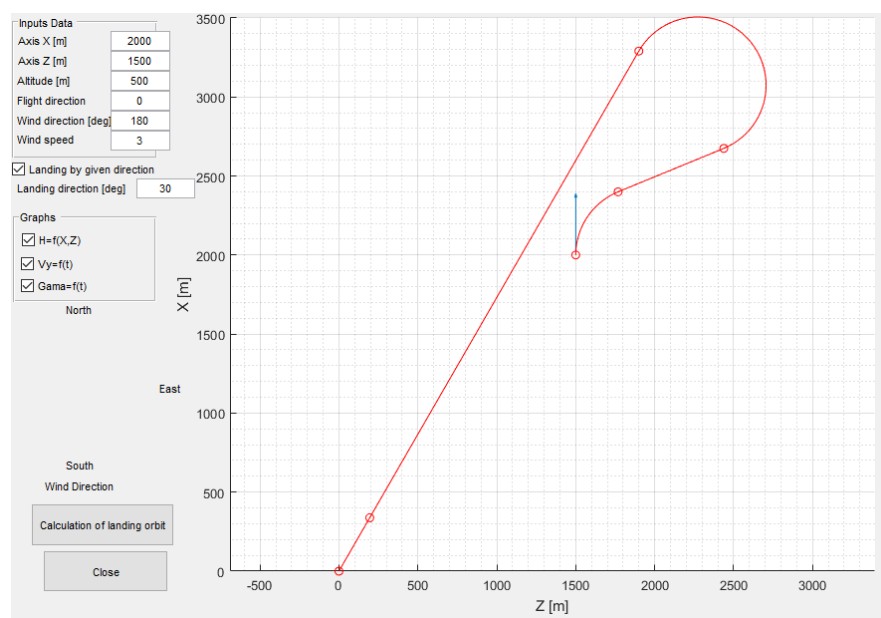

**Figure 22.** The desired trajectory for UAV landing in the given direction.

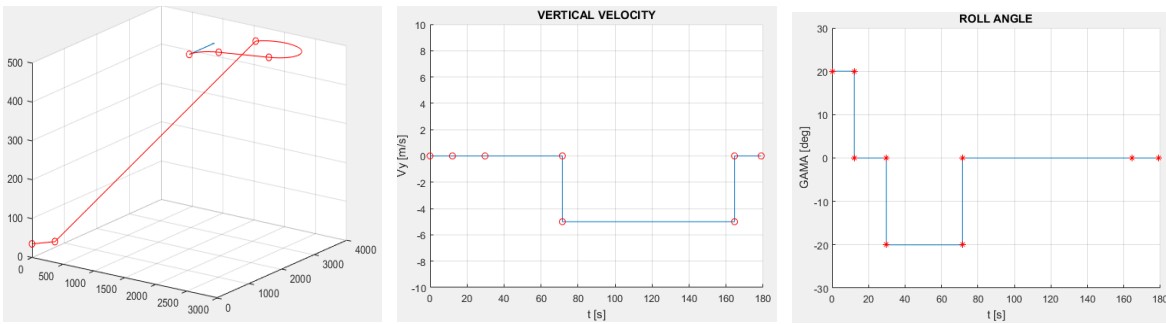

**Figure 23.** The altitude, vertical velocity, and roll angle when the UAV must land in the given direction.

In this case, the desired landing orbit consists of two curves and two lines. At the height H = 500 m, the UAV completes two turning with the desired roll angle $\gamma \leq 20°$. Between these two times, a straight flight takes place with speed V = 40 m/s. Then, the drone flies in the right orbit in the given direction, starting the step of lowering the altitude and finally straight flight at decreasing speed. The simulation result given is reasonable and necessary to implement controlling orders. The result also shows that the landing direction is the direction from the current point of the UAV to the desired landing point, and this landing distance is the shortest one.

## 5. Discussion

We made significant efforts in designing an airway-network for managing drones in traffic flow at smart cities. The proposed new concepts for airway-network operation, such as defining four different sectors and typical elements, are dedicated to the most critical urban air traffic flows. However, due to operational constraints, the complete design of airway-networks needs more investigation and practice.

The pre-defined airway-network constitutes one of the major elements used in urban air traffic management. As air traffic has been concentrated on airway networks, proposed airway networks that have been systematically designed may improve traffic flow and help manage UAVs in urban air traffic flows. The complete design of an airway network is a challenging task that considers many competing factors, such as the needs of both controlled and un-controlled airspaces, the UTM capacity, sectorization scheme, or connectivity to stations.

Although the simulation results of drone-following models can verify the proposed approach, several improvements for drone-following models could be made, including the followings:

- safe distance is measured not only the drone directly in front but also two drones beside;
- outputs of the controller are based on the estimation of the system state at a particular time, which can be used to control the following drones;
- several situations, such as the augmentation or reduction of the number of drones participating in the traffic flow, should be introduced to evaluate more accurately the performance of the SD models as well as the Markov model;
- it is necessary to design and conduct an experimental study to collect quantitative information regarding the drone performance in space, in which one drone cannot pass another.

By this experimental study, statistical estimates of certain functions and parameters for a preliminary evaluation of the mathematical models will be achieved.

The CbDMS is an advanced approach for managing drones to meet critical features. The CbDMS links to Cloud computing, real-time streaming, up-to-date information, and intelligence response to dynamically changing conditions. With CbDMS, complex missions can be taken with ease, enhancing reliability and applicability. However, it can be noted that controlling and managing real-time drones over the network is highly dependent on a reliable quality of service. For example, controlling a drone through the Internet may cause

harm or crash to a drone because of a missing command or have a command with delay. An intelligent onboard system is the best solution to avoid collisions if a command is not received to overcome this problem.

Drones can detect obstacles and plan their paths by using onboard sensors that receive information in real-time. It means that drones can survey and gather environmental information. Keeping this information up to date enables online managing and controlling drones, one of the most advantages of drone applications. However, one drawback may occur in online path designing and obstacle avoidance, such as less accuracy due to insufficient input data. Regarding trajectory and path planning, several criteria, including total travel distance, completion time, coverage area, and maneuvers, are applied to assess the execution of drone applications.

Small UAVs play an essential role in scientific meteorological investigations. The landing process is the most crucial process of the UAVs' flight. To determine the landing areas, solving the system of differential motion of aircraft used, on which the desired landing orbit is calculated. The simulation results show the shapes of the trajectories in different initial conditions. Although the wind is considered, the wind impacts in real-time do not perform in the simulation results. Thus, this method can be developed depending on the actual situations (e.g., wind direction, wind size, humidity) and considering reducing the required landing areas and environmental impact. The improved methods might be applied to the more complex task of landing in city areas or on moving, oscillating platforms.

This work is part of the research that aims to integrate drones with the urban air transportation system. With more insights on the integration of drones into urban air traffic, the study suggests future directions and challenges as follows:

- air network organization and management, which has been the cornerstone for the safe integration of drones. Specifically, air network classification improved in terms of UAV integrated operation in the controlled and uncontrolled airspaces;
- UAV trajectory management: concerning the future advanced operation of concepts, flexible and powerful UAV trajectory management is recommended in the urban air transportation context with guidance and control over the trajectory;
- technology and system improvement. There is still much room for developing communication, control algorithms, and path planning to support efficient, safe, and reliable UAV operations in urban airspace;
- standardization and regulation considerations.

The developed introduced concept and results of verification tests of possible models, methods, and solutions supporting the safe traffic of drones in a smart city environment are compared to the exiting ideas, concept, and available scientific results published. There are two significant groups of novelties that have been introduced.

At first, a developed operational concept was described that has the following specific aspects comparing the available series of the concept of operation of drones applying in urban areas:

- operational concept described here had been developed by using a NextGen and SESAR [11–17] single system operation concept including the total monitoring, that is combined by using the GPS/GNSS positioning [33–35], markers, and active sensors integrated into the infrastructure, too, using the Internet of Things approach [68] hierarchical classification in cooperation, applying the safe separation, sense and avoidance, using the aeronautical information service (AIS) data, and geographic information systems (GIS) data applicable to the UTM airspace design [36,37], GIS structured support and single operational center that all represent the most actual state of the art;
- network of trajectories (tubes and corridors) had been defined as the improved solution of the Singapore recommendation and [18,19] demonstrations [20];
- the introduced trajectory elements and safety criteria are introduced by authors, and that guarantee safe flights of drones in trajectory net without any crossing at same altitude level [69,70];

- the dynamic and active sectorization concept synthesizes the "automated support for dynamic sectorization" developed by SESAR Joint Undertaking megaproject [71] and the concept of airspace configuration [37];
- system defined as a set of six categories of drones (non-detected, detected—not cooperating, etc.) by use of classification used by surveillance and ATM service providers and published by authors in [25,26].

The second part showed and discussed some simulation results of improved solutions, the method needed for supporting the drones' safe traffic, traffic management:

- the sensor fusion had been discussed only;
- desired trajectory following management was tested using a novel system based on a unique inversion-model-based control and test results that result in conclusions. The method can be used in areas between the large houses, in area of possible lack in GPS/GNSS positioning, and accuracy reaches acceptable level even in wind class 5, that have as good accuracy as the available other methods, like [4,6], but in all areas;
- there was created new drone following model based on Markov approximation of the drone following process, that results in the same or slightly better solutions than the widely applied SD model, results show that the Markov model with identified parameters generates more realistic solutions (See Figure 19);
- there was developed a landing management solution the was tested in simulation verification, demonstrating the applicability of the model that has advantages in controlling the landing of the fixed-wing UAV.

## 6. Conclusions

In this paper, we proposed a drone management system in urban areas. To achieve this goal, we first presented operational concepts for drone operations in urban areas, including airspace design, recommended construction of the airways, and essential safety requirements. We then introduced several methods for controlling and managing drone operations, such as the desired trajectory following management, following process, obstacle avoidance, and desired land orbit. Finally, the validation of the proposed method has been satisfied by introducing some developed models as follows:

- drone-following models have been developed to manage drones in urban air traffic flows based on the principle that keeps a safe distance according to relative velocity. The numerical simulation environment demonstrated that the drones' safety distance is maintained; namely, there was no accident in the traffic flow;
- a new managing system for integrating drone motion into urban traffic flow using the cloud-based approach. An improvement of the communication approach allows users to control and monitor drones as connected objects in a real-time environment, which provides the management and control of drone applications for delivery, surveillance, security, ambulance, and emergency response;
- an advanced methodology for determining and calculating UAVs' landing stages was based on the differential system equation of UAV and orbits-straight line trajectory. This method can be applied to the more complex task landing in city areas and moving or oscillating platforms.

By providing such a scenario, we hope to reduce much of the effort currently spent managing a large of UAV, especially drones, with every new autonomous system development project. This will enable researchers, developers, and enthusiasts to develop their autonomous systems with significantly reduced time, effort, and funding. It will also allow professionals to share best practices and best results within the organization and across organizations and disciplines.

A comprehensive discussion about the results, limitations, and future works were presented in this paper. Further research is required to understand better how the proposed system could be planned, which depends on open questions linked to critical features

associated with the possible significant use of drones and how sustainably they could be in life quality.

The authors know well, this paper introduces a developed concept of safe using drones in an urban environment, possible safe integration of the drones' traffic into the smart city transportation system generally. Of course, the implementation of this concept needs further theoretical and practical investigation and applying an extensive series of different methods, techniques, and solutions that were studied, improved, developed by authors, too, as possible examples.

**Author Contributions:** Conceptualization, Jozsef Rohacs and Daniel Rohacs; Data curation, Jozsef Rohacs and Dinh Dung Nguyen; Formal analysis, Jozsef Rohacs and Daniel Rohacs; Investigation, Jozsef Rohacs and Daniel Rohacs; Methodology, Jozsef Rohacs and Dinh Dung Nguyen; Resources, Dinh Dung Nguyen and Daniel Rohacs; Software, Jozsef Rohacs; Supervision, Daniel Rohacs; Validation, Jozsef Rohacs, Dinh Dung Nguyen, and Daniel Rohacs; Writing—original draft, Jozsef Rohacs and Dinh Dung Nguyen; Writing—review and editing, Jozsef Rohacs and Daniel Rohacs. All authors have read and agreed to the published version of the manuscript.

**Funding:** This research was funded by the Hungarian national EFOP-3.6.1-16-2016-00014 project titled by "Investigation and development of the disruptive technologies for e-mobility and their integration into the engineering education."

**Institutional Review Board Statement:** Not applicable. There were not involved humans or animals directly into this studies. However, the developed and recommended system results to lower environmental impact of (drone) traffic and avoiding the accidents, incidents endangering the animals and living nature generally by implementing the sense and avoiding (conflict detection and resolution) system.

**Informed Consent Statement:** Not applicable. There were not involved patients and others into this studies (except researchers creating ideas, making some theoretical, simulation and practical tests with using the drones).

**Data Availability Statement:** In this study there were used data available in cited references, data according to the UAV operated by army and data of the developing drones based on the contracts given by an principal to the department of authors. Therefore some information on data does not available for public, yet.

**Conflicts of Interest:** The author declare no conflict of interest.

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
