# Peer review of "Autonomous Flight Trajectory Control System for Drones in Smart City Traffic Management"

_ijgi, doi:10.3390/ijgi10050338_

Round 1

Reviewer 1 Report

The article deals in its first part with the design of a transport system for drones, where the individual corridors and their limitations are specified, including the sectors of change of direction and height of the drone with regard to safety.

Chapter 2.3.1 recommends safety parameters without justification or background for any research.

The second part describes the sensors and methods that control and implement the trajectory of the drone. It also addresses procedures to reduce drone density and landing.

The results chapter contains graphs for the SD model and Markov model, which, however, are not understandable to the reader. Exactly identical graphs are given in previous works of the authors.

Furthermore, the article presents an experiment of a real drone orbit around four points, where the scale and real deviations of the drone trajectory from the point connectors are not visible. I would expect more testing than just four points.

Figure 21 also does not have a description of the axes and does not show the real height deviation of the drone.

The article proposes certain solutions, which in my opinion are not sufficiently supported by experiments. I would expect the authors of the article to verify the proposed model in a real situation or at least in the laboratory on a reduced scale.

Author Response

The authors would like to thank the reviewer for the comments, which significantly helped to improve the manuscript. We have addressed all the issues raised in the review. For each comment, we provided a response, and also described how the manuscript was revised. Those changes are highlighted within the manuscript. Please see the attached file.

Reviewer 2 Report

This paper is good written and shows a structured approach in  Smart city Traffic management.

My  comment concerns th use of GPS positioning . The authors in abstract, in the Introduction  ,in the Methods and in paragraph 4.2 talk about GPS positioning and no GNSS . Perhaps it would be important explain because they have used obly GPS and not GNSS costellation . The have used RTK ?It would be useful  to provide a more detailed explanation .

The paper is good and need only minor revision

Author Response

(The authors gave the same response as above.)

Reviewer 3 Report

Please find the review report attached.

Author Response

(The authors gave the same response as above.)

Reviewer 4 Report

The article is very interesting and proposes an interesting solution, but requires further research. It is good that such research is made.

Abstract

Abstract is sufficient.

Introduction, Basic idea – supporting materials

The introduction and basic idea is enough explained. But, it is not clearly stated what is the novelty of the paper in reference to other literature.

Methods

Creation of models according to Markov chains is not a novelty attempt. There are more accurate models that can be applied to the presented data in the study. Authors did not include in the study the advantages of acoustic sensors. Also other sensors are not very novelty, and we know its accuracy as well, especially the accuracy of the GPS.

Results

In the study, there is more methods than results. The study needs more investigations and practice. For the present study, it is sufficient. 

Discussion

Discussion is sufficient.

Conclusions

In my opinion, the conclusions section should contain information that this is only experimental system, and more investigation and practice should be performed in order to apply this system.  

Author Response

(The authors gave the same response as above.)

Round 2

Reviewer 1 Report

The article describes the proposed solution, which in my opinion is not sufficiently substantiated by testing and real experiments. From the proposed solution of air traffic, I would expect in particular the incorporation of variable flight parameters such as speed and overall position of the drone, from which possible safety restrictions are derived. In my opinion, this was not fulfilled and I therefore consider the article to be unfinished.

Reviewer 3 Report

The authors have addressed all of my concerns. I have no more comments.